# Study of Competitive Displacement of Curcumin on α-zearalenol Binding to Human Serum Albumin Complex Using Fluorescence Spectroscopy

**DOI:** 10.3390/toxins14090604

**Published:** 2022-08-31

**Authors:** Yifang Li, Hongxia Tan, Hongyuan Zhou, Ting Guo, Ying Zhou, Yuhao Zhang, Xiaozhu Liu, Liang Ma

**Affiliations:** 1College of Food Science, Southwest University, Chongqing 400715, China; 2Chongqing Key Laboratory of Speciality Food Co-Built by Sichuan and Chongqing, Chongqing 400715, China; 3Key Laboratory of Quality and Safety Control of Citrus Fruits, Ministry of Agriculture and Rural Affairs, Southwest University, Chongqing 400712, China; 4Key Laboratory of Luminescence Analysis and Molecular Sensing, Southwest University, Ministry of Education, Chongqing 400715, China; 5Key Laboratory of Condiment Supervision Technology for State Market Regulation, Chongqing 400715, China; 6Foshan Micro Miracles Biotechnology Company, Foshan 528000, China

**Keywords:** α-zearalenol, curcumin, human serum albumin, competitive interaction, intervention, fluorescence spectroscopy

## Abstract

α-zearalenol (α-ZOL) is a mycotoxin with a strong estrogen effect that affects the synthesis and secretion of sex hormones and is transported to target organs through human serum albumin (HSA). Additionally, it has been reported that curcumin can also bind to HSA with high affinity at the same binding site as α-ZOL. Additionally, several studies reported that reducing the bound fraction of α-ZOL contributes to speeding up the elimination rate of α-ZOL to reduce its hazard to organs. Therefore, to explore the influence of a nutrition intervention with curcumin on α-ZOL effects, the competitive displacement of α-ZOL from HSA by curcumin was investigated using spectroscopic techniques, ultrafiltration techniques and HPLC methods. Results show that curcumin and α-ZOL share the same binding site (subdomain IIA) on HSA, and curcumin binds to HSA with a binding constant of 1.12 × 10^5^ M^−1^, which is higher than that of α-ZOL (3.98 × 10^4^ M^−1^). Ultrafiltration studies demonstrated that curcumin could displace α-ZOL from HSA to reduce α-ZOL’s binding fraction. Synchronous fluorescence spectroscopy revealed that curcumin could reduce the hydrophobicity of the microenvironment of an HSA–α-ZOL complex. This study is of great significance for applying curcumin and other highly active foodborne components to interfere with the toxicokinetics of α-ZOL and reduce its risk of its exposure.

## 1. Introduction

α-ZOL is one of the metabolites of zearalenone (ZEN) formed in the metabolism of plants [1], which mainly exists in corn, wheat, rice and other agricultural products [2]. α-ZOL can also be formed through the metabolism of animals after ingesting cereals contaminated with ZEN, which exerts toxic effects on humans and animals [3,4,5,6]. Additionally, Frizzell et al. [7] found that α-ZOL showed a higher estrogen potency (EC_50_ = 0.022 ± 0.001 nM) than ZEN, being approximately 70 times that of ZEN (EC_50_ = 1.6 ± 0.23 nM). In addition, the prolonged exposure of α-ZOL may cause hormone-dependent cancers [8]. In consideration of α-ZOL’s prevalence in cereal grains and its strong estrogen effect on humans and animals, detoxification measures should be developed. At present, most of the studies on the detoxification of α-ZOL are about its effect after reaching target cells or target organs [9,10], but there are few studies on the transport of α-ZOL in vivo, which is of great significance for the detoxification of α-ZOL.

HSA is a major protein in the blood circulation; it is an important carrier protein that can transport endogenous metabolites and exogenous active small molecules to target organs [11,12,13]. Loads of studies [14,15] have reported that mycotoxins can bind to HSA with a high affinity, which could have an influence on the transportation of mycotoxins in vivo. As for α-ZOL, Faisal et al. [14] found that α-ZOL binds to HSA with a high affinity (K_a_ = 5.25 × 10^4^ M^−1^) at HSA’s Sudlow’s site I (subdomain IIA). The tissue distribution and the half-life of α-ZOL could be affected by the formation or destruction of α-ZOL–HSA [16]. Some studies [17] reported that the toxicity of mycotoxin Ochratoxin A (OTA) can be reduced by reducing the half-life of OTA in vivo. It has been reported that some flavonoid aglycones can competitively displace the formation of OTA on HSA and then reduce the half-life of OTA [17]. Additionally, recently, more active substances, such as resveratrol, quercetin and anthocyanin, were confirmed to have the potential to displace Aflatoxin B_1_ from HSA [18,19,20]. Therefore, we speculated that the interaction between α-ZOL and HSA can be affected by other substances in in vivo circulation, which could reduce the binding constant of α-ZOL and HSA. Curcumin (Cur) has been reported to have a high affinity with HSA and bind to HSA at the same site as that of α-ZOL (subdomain IIA) [21]. Additionally, Maciążek-Jurczyk et al. [21] have found that curcumin has the ability to affect the binding fraction of the Tamoxifen–HSA system. Curcumin has great anti-inflammatory properties, an antioxidant capacity and the ability to inhibit mycotoxin toxicity [22,23], so we suggest that curcumin is a possible competitor for α-ZOL binding with HSA.

For the sake of reducing the half-life of α-ZOL, and alleviating the toxicity caused by α-ZOL on target organs or cells, the competitive replacement of curcumin for α-ZOL from HSA was studied under the blood pH (7.4) environment in vitro using fluorescence spectra methods and ultrafiltration and HPLC studies.

## 2. Results

### 2.1. The Comparison of the Affinity of α-ZOL–HSA and Curcumin–HSA Complexes 

We investigated the binding mechanism of a curcumin–HSA complex and α-ZOL–HSA complex by observing the fluorescence spectra of the curcumin–HSA complex and α-ZOL–HSA complex. As shown in Figure 1A,B, the endogenous fluorescence of HSA could be quenched by both curcumin and α-ZOL at 298 K, and the peak position had no obvious shift. The binding mechanism was obtained by Equations (1) and (2). As shown in Table 1, at 298 K, the K_q_ of the α-ZOL–HSA complex and curcumin–HSA complex was far greater than the maximum collision quenching constant (2.0 × 10^10^ L/mol∙s) [24], indicating that the quenching mechanism of the α-ZOL–HSA complex and curcumin–HSA complex was static quenching. For the static quenching process, the fluorescence quenching double logarithmic curve can be made according to the Lineweaver–Burk static quenching Formula (3). Thus, the binding constant K_a_ and the numbers of corresponding binding sites n can be calculated. As shown in Table 1, both curcumin and α-ZOL had one binding site in HSA, and the binding constant K_a_ of the curcumin–HSA complex was one order of magnitude higher than that of the α-ZOL–HSA complex, which indicates that curcumin has a higher affinity for HSA than α-ZOL.

### 2.2. Analysis of the Site of Curcumin–α-ZOL on HSA

There are two main binding sites on HSA that can bind with ligands: Sudlow’s site I (located in subdomain IIA) and Sudlow’s site II (located in subdomain IIIA). In order to further confirm the binding site of curcumin and α-ZOL on HSA, warfarin and ibuprofen were used, which are the markers of HSA on site I and site II, respectively [25,26]. Figure 1C,D, show that the fluorescence was noticeably decreased by the gradual addition of warfarin. However, the addition of ibuprofen had a minor effect on fluorescence spectra of HSA. This result indicates that α-ZOL and curcumin mainly competed with warfarin and were most likely bound to Sudlow’s site I (subdomain IIA) of HSA [27], which is consistent with previous studies [14,21].

### 2.3. Effect of Curcumin on α-ZOL–HSA Interaction 

According to the fluorescence spectra results of HSA–α-ZOL and HSA–curcumin complexes, curcumin and α-ZOL can bind to HSA at the same site. Moreover, curcumin bound to HSA with a higher affinity than α-ZOL. Thus, we suggest that curcumin is able to interfere with the interaction of the HSA–α-ZOL complex. To investigate the effect of curcumin on the HSA–α-ZOL complex, we studied the fluorescence spectra of the HSA–α-ZOL–curcumin complex [19]. From Figure 2A, it was clear that even though the HSA–α-ZOL complex had been formed, the endogenous fluorescence of HSA could still be uniformly quenched with the increase in curcumin concentrations. The K_sv_, K_q_ and K_a_ of the curcumin–HSA complex were calculated by Equations (2) and (3). As shown in Table 2, in the presence of α-ZOL, the quenching mechanism of curcumin on HSA was still static quenching, and α-ZOL did not change the quenching type. Although the binding constant of the curcumin–HSA complex had changed, it was still greater than that of the HSA–α-ZOL complex. It can be seen from Figure 2B that in the ternary system, curcumin and α-ZOL were still combined at site I of HSA, indicating that curcumin and α-ZOL will compete for binding site I.

### 2.4. Studies of Competive Displacement of Curcumin on α-ZOL from HSA by Ultrafiltration–HPLC Studies

In order to study whether curcumin can replace the α-ZOL from HSA to form the HSA–curcumin complex, instead of forming an HSA–α-ZOL–curcumin complex, an ultrafiltration experiment was carried out. After the equilibration (incubation) of curcumin with HSA–α-ZOL mixtures according to Section 5.4, we separated the free α-ZOL (unbound fraction) and the HSA–α-ZOL complex (bound fraction) by ultrafiltration [28]. Therefore, the concentration of α-ZOL in the free state was examined by HPLC to confirm whether curcumin was able to displace α-ZOL from HSA, increasing the rate of α-ZOL in filtrate. 

The HPLC chromatograms of free α-ZOL in different systems are presented in Figure 2C, and the concentration of free α-ZOL is shown in Figure 2D. We can see that with the addition of the ratio of α-ZOL–curcumin from 1:0 to 1:9, there was a significant difference (*p* < 0.05) in the concentration of free α-ZOL among different systems. With the increasing rates of curcumin, the concentration of α-ZOL in the free state gradually increased, indicating that curcumin can displace the α-ZOL in the HSA–α-ZOL complex. 

### 2.5. Effect of the Competitive Replacement of Curcumin on α-ZOL on Microenvironment of HSA

Synchronous fluorescence spectroscopy is an effective approach to investigate the microenvironment around protein chromophores [29]. For the synchronous fluorescence spectrum, the wavelength interval between excitation and emission was fixed at 15 nm and 60 nm to obtain the information regarding conformational or polarity variation around tyrosine (Tyr) and tryptophan (Trp) residues, respectively [30,31]. The synchronous fluorescence spectroscopy of HSA–α-ZOL at Δλ = 15 nm and Δλ = 60 nm is shown in Figure 3A,B. We found that in the presence of α-ZOL, a blue shift occurred in the maximum emission peak of both Tyr and Trp residues, which suggests that α-ZOL–HSA interaction increased the hydrophobicity and decreased the polarity of the microenvironment of Tyr and Trp. The effects of curcumin on the synchronous fluorescence spectroscopy of the HSA–α-ZOL complex at Δλ = 15 nm and Δλ = 60 nm are shown in Figure 3C,D. As the concentration of curcumin increased, the maximum emission wavelength had a slight red shift of 0.2 nm at Δλ = 15 nm, and a red shift of 1.8 nm at Δλ = 60 nm. The above results suggest that in ternary system, curcumin could reduce the hydrophobicity of the protein microenvironment to affect the interaction of α-ZOL with HSA.

### 2.6. Effect of the Ratio of HS–α-ZOL–Curcumin on the Competitive Replacement

To evaluate the competitive replacement of curcumin for α-ZOL, the effect of the ratio of HSA–α-ZOL–curcumin on the fluorescence intensity of HSA was analyzed. In Section 2.1, it was found that both curcumin and α-ZOL can bind with HSA at site I with a 1:1 stoichiometry within the range of the investigated concentrations. Moreover, the ultrafiltration study in Section 2.4 confirmed that the fluorescence intensity changes in the HSA–α-ZOL complex were due to the displacement of α-ZOL by curcumin and formation of the HSA–curcumin complex. In this section, a complex of HSA–α-ZOL with a ratio of 1:1 was firstly obtained, and then the effect of curcumin on the HSA–α-ZOL complex was studied. As shown in Figure 4A, there was a significant difference in the fluorescence intensity of HSA between HSA–α-ZOL and HSA–α-ZOL–curcumin systems. With the increasing ratio of curcumin from 1:1:0.5 to 1:1:2, the fluorescence of HSA significantly decreased (*p* < 0.05), which indicates that curcumin could displace α-ZOL, even at low ratio of 1:1:0.5. However, no significant effects of α-ZOL on the HSA–curcumin system could be observed. With the addition of α-ZOL to the HSA–curcumin system, there was no significant difference (*p* < 0.05) in the fluorescence intensity of HSA between HSA–curcumin and HSA–curcumin–α-ZOL systems until the concentration of α-ZOL was 2 times higher than that of curcumin. These results confirm that the rate of α-ZOL binding to HSA was lower than that of curcumin binding to HSA. 

The concentration of α-ZOL that humans take in vivo is a low level, which is far less than the concentration of HSA in humans; thus, the effects of the curcumin on the HSA–α-ZOL complex and those of α-ZOL on HSA–curcumin complex were also analyzed at the low ratio of 1:0.5 between HSA and ligands. As shown in Figure 4B, there was a significant difference (*p* < 0.05) in the fluorescence intensity of HSA between the HSA–α-ZOL–curcumin systems (1:0.5:0.5 and 1:0.5:1) and the HSA–α-ZOL system (1:0.5). Meanwhile, there was no significant difference (*p* < 0.05) in the fluorescence intensity of HSA between the HSA–curcumin system (1:0.5) and HSA–curcumin–α-ZOL (1:0.5:0.5). These results are consistent with the results shown in Figure 4A, further confirming that curcumin has a higher affinity with HSA than α-ZOL. 

Additionally, the effects of the curcumin on the HSA–α-ZOL complex and those of α-ZOL on the HSA–curcumin complex were also further analyzed at the high ratio of 1:2 between HSA and ligands. As shown in Figure 4C, the addition of curcumin to the HSA–α-ZOL system could significantly decrease the fluorescence of HSA, whereas the addition of α-ZOL to HSA–curcumin did not significantly affect the fluorescence of HSA. 

Based on the above results, we found that compared with the α-ZOL, curcumin had a stronger binding affinity with HSA, and is thus capable of replacing the α-ZOL from HSA. Furthermore, whether the ratio of HSA and α-ZOL was 1:0.5, 1:1 or 1:2, the addition of curcumin could displace α-ZOL from HSA. The more curcumin added, the more obvious the competitive substitution for α-ZOL from HSA.

## 3. Discussion

The mycotoxins–albumin binding strongly affects the toxicokinetics of mycotoxins. Reducing the bound fraction of mycotoxin could shorten the half-life of circulating mycotoxin. Previous studies [32,33,34,35,36] showed that mycotoxins mainly bind to HSA with a high albumin affinity (K_a_~10^7^ L mol^−1^), which results in the longer elimination half-life of mycotoxins, preventing excretion by liver and kidney metabolism. It has been found that reducing the bound fraction of mycotoxins with HSA can be used to decrease its half-life, which was reported by Kumagai et al. [36], who found that the excretion rate of OTA in albumin-deficient rats was 20–70 times higher than in normal rats when OTA was injected. Therefore, we can speculate that several compounds capable of displacing mycotoxins from serum albumin are expected to be efficient antidotes.

The nutritional intervention effect of foodborne components (e.g., polyphenols) are of particular concern. Previous studies have shown that many substances have the ability to intervene or affect the toxicity of mycotoxins by reducing the binding of the mycotoxins–HSA complex. TAN et al. [18] studied the competitive displacement of quercetin for aflatoxin B_1_ (AFB_1_) from HSA, which found that quercetin can remove AFB_1_ from HSA. Qureshi et al. [20] also found that the binding constant between resveratrol and HSA was as high as 10^7^, with the potential to remove AFB_1_ from HSA. Additionally, many studies have reported that the displacement between the drugs and albumin is a feasible approach to provide safer therapies and achieve particular therapeutic goals [16].

Based on this, to alleviate the hazard of α-ZOL to organisms, the competitive displacement of α-ZOL from human serum albumin by curcumin was studied in this paper. Experimental results of fluorescence spectroscopy (Section 2.1, Section 2.2 and Section 2.3) show that curcumin has the potential to replace α-ZOL in HSA. Using ultrafiltration studies (Section 2.4), it was further confirmed that curcumin could remove α-ZOL from HSA and then decrease the bond fraction of α-ZOL. Similar results for competitive binding were obtained for Tamoxifen. Maciążek-Jurczyk et al. [21] studied competitive binding of tamoxifen with curcumin, and found that when curcumin is present in the Tamoxifen–HSA system, the interaction of Tamoxifen–HSA is more difficult or hinders the binding of Tamoxifen to HSA. Curcumin is a hydrophobic substance, and the structurally active groups are mainly phenolic hydroxyl groups and methoxy groups [37]. Through molecular docking research, SHAIKH et al. [38] found that Gln204 and Thr243 on HSA are involved in the electrostatic interaction of methoxy with hydroxyl groups of curcumin, respectively. Additionally, amino acid residues are involved in the hydrophobic effect of curcumin aromatic groups and enone backbones. Based on the synchronous fluorescence spectroscopy results (Figure 3), we suggest that in the HSA–α-ZOL complex, there is a hydrophobic interaction between curcumin and HSA, and the phenolic hydroxyl groups and methoxy groups of curcumin interact with the amino acid residues on HSA, thereby binding to site I of HSA to form a more stable complex. It reduced the binding rate of α-ZOL, resulting in an increase in the free fraction of α-ZOL and reduction in the half-time of α-ZOL, which contributes to reducing the toxic effects on humans and animals. The results of this paper show that a health intervention with curcumin to reduce the toxic effects of α-ZOL is a feasible idea. 

Curcumin is a low-molecular-weight and lipophilic natural polyphenolic compound that is primarily used as a food coloring additive. It has a wide range of pharmacological activities that could be applied as health food [39]. However, these results have not been reflected well mainly due to curcumin’s low water solubility and chemical instability, which results in the low oral bioavailability of curcumin. In recent years, there has been significant research with the purpose of increasing its bioavailability, such as formation of emulsions [40], cyclodextrin complexes [41], nanosuspensions by antisolvent precipitation [42] and polysaccharide complexes [43]. Ubeyitogullari et al. [44] described a novel nanomanufacturing method utilizing supercritical carbon dioxide (SC-CO_2_) technology and nanoporous starch aerogels (NSAs) to improve the bioavailability of curcumin. Curcumin nanoparticles significantly enhanced the bioaccessibility of curcumin by 173-fold compared to the original curcumin. Somu et al. [45] found the cytotoxicity of nanoformulated curcumin (NSCS-Cur) was higher in four different cancer cells (breast cancer; MCF-7 and MDAMB231, cervical cancer; HeLa, osteosarcoma; MG 63) than that of free curcumin. These approaches have the potential to enhance the efficacy of curcumin and allow food manufacturers to incorporate curcumin into foods as a nutritional intervention for humans and animals.

## 4. Conclusions

The effect of curcumin on the HSA–α-ZOL complex was explored by fluorescence spectroscopy and ultrafiltration–HPLC studies. Fluorescence spectra and ultrafiltration study showed that curcumin can replace α-ZOL in the HSA–α-ZOL complex to reduce the binding fraction of α-ZOL. Furthermore, the addition of curcumin can replace the α-ZOL from HSA, regardless of the ratio of HAS with α-ZOL (e.g., 1:0.5, 1:1 or 1:2). This study provided an important reference value for exploring the nutritional intervention effects of curcumin on α-ZOL. Furthermore, in the future, in vivo experiments on the competitive displacement of curcumin by α-ZOL from HSA are desirable.

## 5. Materials and Methods

### 5.1. Reagents

α-Zearalenol (α-ZOL, purity ≥ 98%, Pribolab, Singapore); Curcumin (CUR, purity ≥98%, Victory Biological, Sichuan, China); Human Serum Albumin (Sigma, St. Louis, MO, USA); Tris (Bio Basic, Amherst, NY, USA); Warfarin (purity ≥ 97%, Adamas-beta, Beijing, China) and Ibuprofen (purity ≥ 97%, J&K Scientific, Beijing, China) were obtained. The water used in this study was all ultrapure water, and all the reagents were analytical-grade. 

### 5.2. Fluorescence Spectroscopy

The fluorescence experiments were performed using an F-2500 fluorescence spectrophotometer (Hitachi, Tokyo, Japan) at 298 K with a 1.0 cm-path-length quartz cell. Fluorescence detection conditions were as follows: λ_ex_ = 280 nm, emission wavelength of 300~500 nm, and scanning speed was 240 nm/min. In the HSA–α-ZOL and HSA–curcumin systems, the concentration of HSA was fixed at 10 µM, and the α-ZOL and curcumin concentrations ranged from 0 to 12 µM, respectively. As for the HSA–α-ZOL–curcumin system, we first added α-ZOL to the HSA complex, forming the α-ZOL–HSA complex. Then, concentrations of curcumin ranging from 0 to 12 µM were added to the HSA–α-ZOL complex, in which the concentration of HSA was fixed at 10 µM, and the α-ZOL concentration was 2 µM.

To eliminate the inner-filter effects, fluorescence data were corrected using the following equation [19]:(1)$$F_{\mathrm{cor}}{=F}_{\mathrm{obs}}{\times e}^{\frac{A_{\mathrm{ex}}{+A}_{\mathrm{em}}}{2}}$$

F_cor_ represents the corrected fluorescence intensities and F_obs_ represents the observed fluorescence intensities, whereas A_ex_ and A_em_ are the absorbances of the ligands at excitation and emission wavelength of HSA, respectively.

The quenching rate constant K_q_ can be calculated according to the Stern–Volmer equation [46,47]:(2)$$\frac{F_{0}}{F}{=1+K}_{q}\tau_{0}\left[Q\right]{=1+K}_{\mathrm{sv}}\left[Q\right]$$
where F_0_ and F represent the fluorescence intensity of HSA in the absence and presence of a quencher, respectively. K_q_ is the quenching rate of the biomolecule (L/mol·s). τ_0_ is the average fluorescence lifetime of HSA without a quencher. for HSA, its value is 10^−8^ s; K_sv_ is the Stern–Volmer quenching constant (L/mol). [Q] is the concentration of the quencher (mol/L).

As for static quenching, the binding constant (K_a_) and number of binding sites (n) of the ligand can be calculated using the Lineweaver–Burk equation [15,48]: (3)$$\mathrm{Log}\frac{{(F}_{0}-F)}{F}{=\mathrm{logK}}_{a}+\mathrm{nlog}[Q]$$
where log[(F_0_ − F)/F] denotes the ordinate coordinate and log[Q] denotes the abscissa. The double logarithmic graph fit can be calculated according to the intercept and slope. The binding constant K_a_ and the number of binding sites n can be calculated.

### 5.3. Competitive Probe Studies

Site competition experiments: warfarin (as the site I marker) and ibuprofen (as the site II marker) were used separately. The concentration of HSA was set at 2 µM, and α-ZOL and curcumin were added separately. The ratio of HSA to curcumin/α-ZOL was 1:2. Then, warfarin/ibuprofen with a concentration range of 0~12.0 μM was added to the complexes.

The percentage of α-ZOL and curcumin being substituted with the site markers was calculated by Equation [18].
(4)$$\mathrm{Displcement}=\frac{F_{2}}{F_{1}}$$

F_1_ represents the fluorescence intensity of the HSA–ligand binding system without a specific binding site marker, and F_2_ represents the fluorescence intensity of the HSA–ligand binding system in the absence of a specific binding site marker.

### 5.4. Ultrafiltration–HPLC Study

The prepared α-ZOL was added to each centrifuge tube. The concentration ratio of α-ZOL–HSA was 1:1, and the complex was mixed thoroughly using a QL 901 Vortex Mixer (Kylin-Bell Lab Instruments, Haimen, China) and reacted for 20 min at room temperature (298 K). Different concentrations of curcumin were then added to the α-ZOL–HSA complex, forming the HSA–α-ZOL–curcumin complexes with the different ratios (including 1:1:1, 1:1:3, 1:1:5 and 1:1:9). The separation of free α-ZOL with the α-ZOL–HSA–curcumin complex was carried out through a 10 kDa ultrafiltration tube, which was centrifugated at 10,000 r/min at 4 °C for 15 min [18]. The centrifugation operation was repeated at least 3 times. For each round of centrifugation, the mixed solution of methanol–water (2:98; *v*/*v*) was added to the ultrafiltration tube to elute the free α-ZOL, and then the filtrated solution was collected, followed by drying by nitrogen. Finally, the α-ZOL was re-dissolved with 1 mL of mobile phase (methanol–water; 65:35) and detected by HPLC. The HPLC analysis was performed using the Dionex Ultimate 3000 RS HPLC system (Thermo Fisher Scientific, Waltham, MA, USA), and the chromatographic separation was performed at 25 °C on a reversed-phase HPLC analytical column, 150 mm × 4.6 mm, 5 μm (Agilent, Santa Clara, CA, USA). To detect α-ZOL, samples were eluted at the flow rate of 0.800 mL/min with an injection volume of 100 μL, and the fluorescence detection condition was λ_ex_ = 315 nm, λ_em_ = 455 nm. Retention time for each sample was 20 min.

### 5.5. Synchronous Fluorescence Spectra

The synchronous fluorescence spectra were examined using an F-2500 fluorescence spectrophotometer (Hitachi, Tokyo, Japan) at room temperature (298 K). The concentration of HSA was set at 2 µM in the HSA–α-ZOL system, and the concentration of α-ZOL ranged from 0 to 12 µM. In the HSA–α-ZOL–curcumin system, α-ZOL was first added to the HSA solution at the concentration of 2 µM to form the α-ZOL–HSA = 1:1 complex, and then curcumin was added to the α-ZOL–HSA complex with a concentration varying from 0 to 12 µM. The wavelength intervals between excitation and emission were fixed at 15 nm and 60 nm, respectively.

### 5.6. Effect of Concentration Ratio on Competitive Response

Detection of fluorescence intensity of HSA in different systems was performed, in which the concentration ratio of α-ZOL–curcumin was 1:2, 1:1 and 2:1, respectively. Fluorescence spectroscopy conditions were consistent with Section 5.2. The concentration of HSA was fixed at 2 µM, and the concentration of α-ZOL and curcumin followed the ratio in Figure 4. Additionally, different complexes were built as follows: The HSA–α-ZOL complex was constructed by adding α-ZOL to the HSA solution. The method for constructing the HSA–curcumin complex is similar to that used for the HSA–α-ZOL complex. As for the HSA–α-ZOL–curcumin complex, we first added α-ZOL to the HSA solution, then curcumin was added to the solution. For the HSA–curcumin–α-ZOL complex, curcumin was first added to the HSA solution, and then α-ZOL was added to the solution.

### 5.7. Data Analysis

All data were statistically analyzed using Microsoft Excel 2016 (Microsoft, Redmond, WA, USA) and SPSS Statistics 20.0 (IBM, Armonk, NY, USA). The significance analysis was performed using one-way ANOVA, and fluorescence scan maps were plotted with Origin 2019b (OriginLab, Northampton, MA, USA).

## Figures and Tables

**Figure 1 toxins-14-00604-f001:**
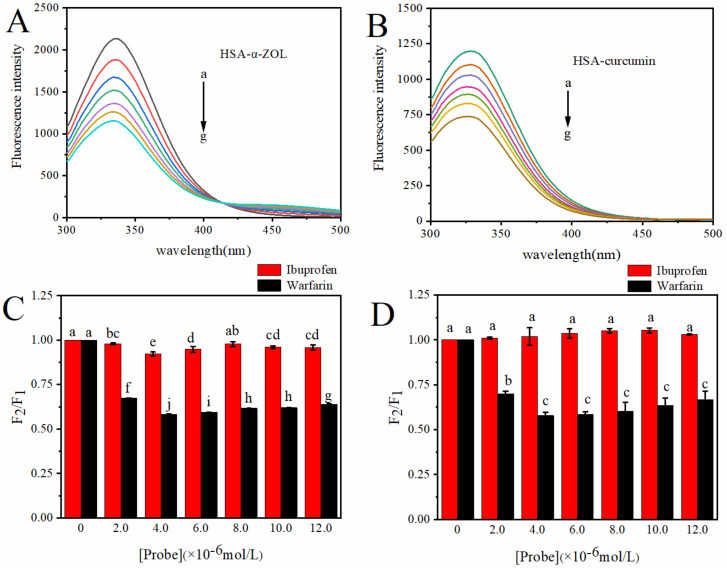
(**A**) Fluorescence emission spectra of HSA (10 µM) in the presence of increasing α-ZOL concentration. (**B**) Fluorescence emission spectra of HSA (10 µM) in the presence of increasing curcumin concentration. C(α-ZOL) = C(curcumin) = 0, 2, 4, 6, 8, 10, 12 µM from a–g. Fluorescence spectra conditions: T = 298 K, pH = 7.4, λ_ex_ = 280 nm. (**C**) Effects of probes on fluorescence intensity of HSA in HSA–α-ZOL complex. (**D**) Effects of probes on fluorescence intensity of HSA in HSA–curcumin complex. Different characters show significant differences (*p* < 0.05). F_1_: fluorescence intensity of HSA when the probes were not added to the solution. F_2_: fluorescence intensity of HSA when the probes were added to the solution.

**Figure 2 toxins-14-00604-f002:**
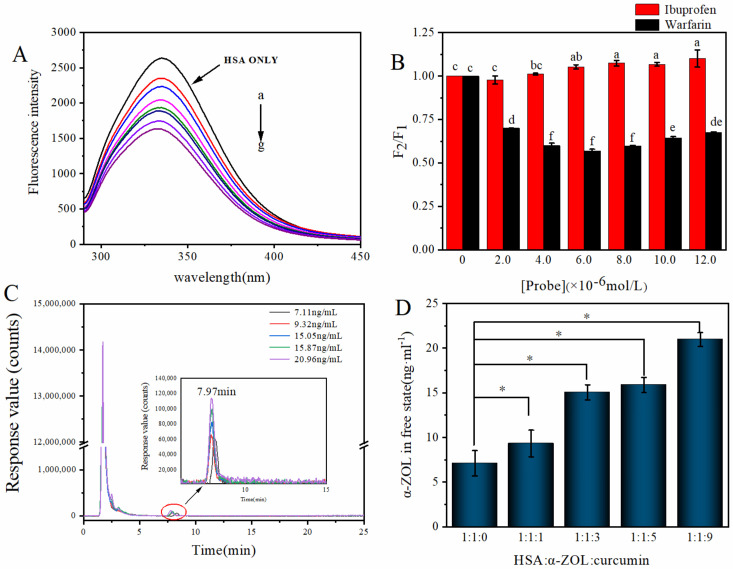
(**A**) Effect of curcumin on the fluorescence spectra of α-ZOL–HSA. C(HSA) = 10 µM, C(α-ZOL) = 2 µM; C(curcumin) = 0, 2, 4, 6, 8, 10, 12 µM from a-g. (**B**) Effects of probes on fluorescence intensity of HSA in α-ZOL–HSA–curcumin complex. Different characters show significant differences (*p* < 0.05). (**C**) HPLC chromatograms of free α-ZOL in different systems. (**D**) Concentration of α-ZOL in the free state in different systems. Different systems mean that different concentrations of curcumin were added to α-ZOL–HSA (1:1) complex, forming the α-ZOL–HSA–curcumin systems with different ratios. *: It shows the significant difference among data (*p* < 0.05).

**Figure 3 toxins-14-00604-f003:**
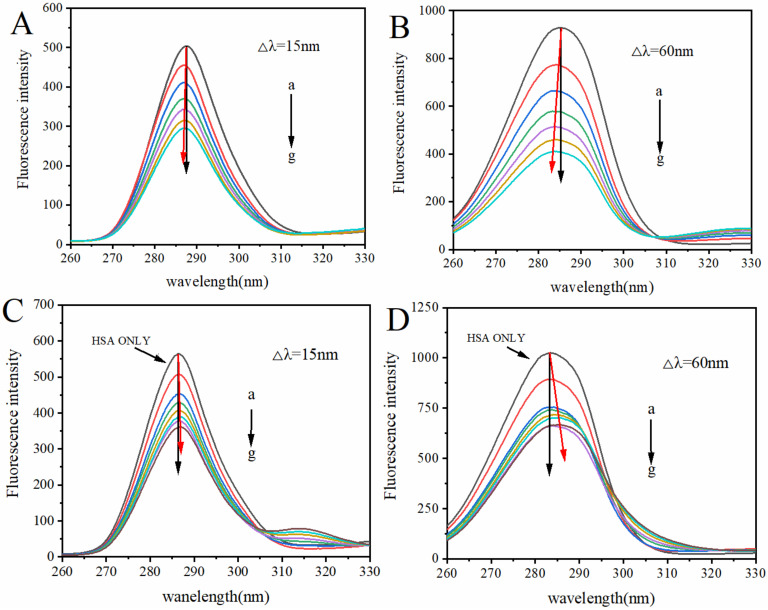
(**A**) Synchronous fluorescence spectroscopy of HSA–α-ZOL system at Δλ = 15 nm; (**B**) synchronous fluorescence spectroscopy of HSA–α-ZOL system at Δλ = 60 nm. C(HSA) = 2 µM, C(α-ZOL) = 0, 2, 4, 6, 8, 10, 12 µM from a–g. (**C**) Synchronous fluorescence spectroscopy of curcumin–HSA–α-ZOL system at Δλ = 15 nm; (**D**) synchronous fluorescence spectroscopy of curcumin–HSA–α-ZOL system at Δλ = 60 nm. C(HSA) = 2 µM, C(α-ZOL) = 2 µM; C(curcumin) = 0, 2, 4, 6, 8, 10, 12 µM from a–g. The red arrows represent the shift in the maximum emission peak of both Tyr and Trp residues.

**Figure 4 toxins-14-00604-f004:**
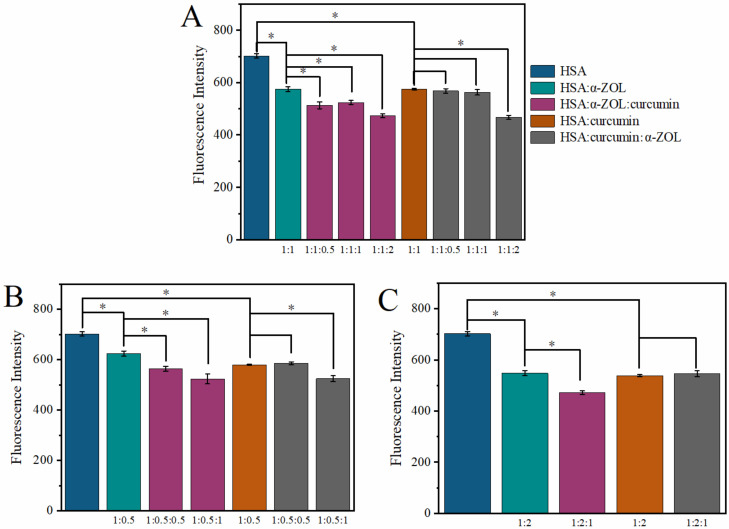
(**A**) The effect of the curcumin on HSA–α-ZOL (1:1) complex and that of α-ZOL on HSA–curcumin (1:1) complex. (**B**) The effect of the curcumin on HSA–α-ZOL (1:0.5) complex and that of α-ZOL on HSA–curcumin (1:0.5) complex. (**C**) The effect of the curcumin on HSA–α-ZOL (1:2) complex and that of α-ZOL on HSA–curcumin (1:2) complex. The concentration of HSA was fixed at 2 µM, and the concentrations of α-ZOL and curcumin follow the ratio in the figure. *: It represents the existence of statistical significance (*p* < 0.05).

**Table 1 toxins-14-00604-t001:** Quenching constant K_sv_, quenching rate constant K_q_, binding constant K_a_ and the possible number of binding sites (n) for curcumin–HSA complex and α-ZOL–HSA complex.

Complex	K_sv_ (×10^4^ L moL^−1^)	K_q_ (×10^12^ L moL^−1^s^−1^)	R	K_a_ (M^−1^)	n	R
curcumin–HSA	2.92	2.92	0.966	1.12×10^5^	1.13	0.980
α-ZOL–HSA	7.15	7.15	0.999	3.98×10^4^	0.98	0.994

**Table 2 toxins-14-00604-t002:** Quenching constant K_sv_, quenching rate constant K_q_, binding constant K_a_ and the possible number of binding sites (n) for HSA–curcumin systems.

C_α-ZOL_ (×10^−6^ L moL^−1^)	K_sv_ (×10^4^ L moL^−1^)	K_q_ (×10^12^ L moL^−1^s^−1^)	R	K_a_ (M^−1^)	n	R
0	2.92	2.92	0.966	11.20	1.13	0.980
2.0	3.64	3.64	0.991	14.40	1.12	0.990
4.0	3.84	3.84	0.989	8.32	1.07	0.982

## Data Availability

The datasets generated for this study are available on request from the corresponding author.

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
