# Peer review of "Study of Competitive Displacement of Curcumin on α-zearalenol Binding to Human Serum Albumin Complex Using Fluorescence Spectroscopy"

_toxins, 2022, doi:10.3390/toxins14090604_

Round 1

Reviewer 1 Report

The study focuses on investigating the competitive replacement of mycotoxin (a-ZOL) from HSA by curcumin. The authors demonstrated that the both  a-ZOL and curcumin binds at the same site on HSA that is Sudlow's site I. Furthermore, the author demonstrated that the binding affinity of curcumin is higher than a-ZOL and can replace the aZOL from HSA-a-ZOL complex even at lower concentrations. The study is interesting and the experiments were well designed and executed. Here are few comments for the authors:

a) It would be interesting to include a figure similar to Figure 2A demonstrating the effect of a-ZOL on the fluorescence spectra of curcumin-HSA. It will eliminate all the possibilities that the a-ZOL has higher affinity and can replace curcumin.

b) Line 126 What is "in different systems"?

c) Line 168 - 171, I disagree with authors that a-ZOL is not capable of displacing curcumin from HSA. In figure 4A, HSA:curcumin:a-ZOL (1:1:2) clearly has some effect on fluorescence indicating replacement of curcumin by a-ZOL (same could be seen in Figure 4B, alythough minimally).

d) There are few minor spelling and grammatical errors that authors should correct while proof reading. Eg., Line 188 "We can find" is not appropriate, Line 156 "the effect the ratio" etc.

Author Response

Dear Reviewer:

According to your comments, a drastic revision of my manuscript has been done. All of the changes have been marked in the revised manuscript in red. The format, spelling, language, information of authors in manuscript have also been revised. The details are as followings.

I marked these changes using “Track Changes” function and colored red.

Reviewer #1:

The study focuses on investigating the competitive replacement of mycotoxin (α-ZOL) from HSA by curcumin. The authors demonstrated that the both α-ZOL and curcumin binds at the same site on HSA that is Sudlow's site I. Furthermore, the author demonstrated that the binding affinity of curcumin is higher than α-ZOL and can replace the a-ZOL from HSA-α-ZOL complex even at lower concentrations. The study is interesting and the experiments were well designed and executed. Here are few comments for the authors:

1) It would be interesting to include a figure similar to Figure 2A demonstrating the effect of α-ZOL on the fluorescence spectra of curcumin-HSA. It will eliminate all the possibilities that the α-ZOL has higher affinity and can replace curcumin.

Thanks for your review. The effect of α-ZOL on the fluorescence spectra of curcumin-HSA have been displayed in section 2.6. In this section, the results showed that compared with curcumin, α-ZOL had a lower affinity to replace curcumin.

2) Line 126 What is "in different systems"?

Thank you for your comments. The “different systems” means that different concentrations of curcumin were added to α-ZOL: HSA=1:1 complex, forming the different ratio of α-ZOL-HSA-curcumin systems. We have added more detailed explanations in the caption of Figure 2 in Line136-139.

3) Line 168 - 171, I disagree with authors that α-ZOL is not capable of displacing curcumin from HSA. In figure 4A, HSA:curcumin:α-ZOL (1:1:2) clearly has some effect on fluorescence indicating replacement of curcumin by α-ZOL (same could be seen in Figure 4B, although minimally).

Thanks for your suggestions. We agree with your suggestion. We have changed it in revised manuscript in Line 196-198.

4) There are few minor spelling and grammatical errors that authors should correct while proof reading. Eg., Line 188 "We can find" is not appropriate, Line 156 "the effect the ratio" etc.

Thanks for your suggestions. We have corrected some words and grammars in revised manuscript.

In Line 215 "We can find" has been revised as “We found”, and in Line 181 "the effect the ratio" has been revised as "the effect of the ratio"

Reviewer 2 Report

The authors have done good work on this interesting research and prepared a well-written manuscript. The manuscript is interesting for the readers, the research design is appropriate, and the methods are adequately described. All the cited references are relevant to the research, however, the introduction section should provide more background on the related research and include the relevant references on this research.

Results are clearly presented, but the conclusions could be more supported by the results- in terms of including the main research outputs.

Author Response

Dear Reviewer:

According to your comments, a drastic revision of my manuscript has been done. All of the changes have been marked in the revised manuscript in red. The format, spelling, language, information of authors in manuscript have also been revised. The details are as followings.

I marked these changes using “Track Changes” function and colored red.

Reviewer #2:

The authors have done good work on this interesting research and prepared a well-written manuscript. The manuscript is interesting for the readers, the research design is appropriate, and the methods are adequately described. All the cited references are relevant to the research; however, the introduction section should provide more background on the related research and include the relevant references on this research.

Results are clearly presented, but the conclusions could be more supported by the results-in terms of including the main research outputs.

Thank you for your comments. In introduction, we have made some changes on this point in our revised manuscript in Line 53-56, 61-63 and 67-70. In conclusion, we have added the outputs to help support the conclusions in Line 295-306.

Reviewer 3 Report

The authors describe the ability of curcumin to displace alpha-zearalenol from HSA and, thereby, reducing the half life of alpha-zearalenol and its detrimental effects. The described study is novel and performed well enough. Unfortunately, the manuscript suffers from poor English. As an example, the sentence starting on line 46 "Therefore, we speculated that the interaction between a-ZOL and HSA can be interfered by other substances in vivo circulation, which could reduce the binding constant of a-ZOL and HSA." This sentence should be present tense, not "interfered by" but affected by, and not the binding constant is reduced but the amount of bound a-ZOL. There are a number of more locations that need revision.

L69: Should this not be "number of binding sites n" instead of "binding site n"?

L73ff: the abbreviations in the column headers need to be defined. There must be a space between L and mol and a lower case s.

L251: Abbreviations "SC-CO2" and "NSAs" used without definition.

L272: Sources of the chemicals need to be provided.

L315: Details about the HPLC method to determine free a-ZOL are missing. Description of the ultrafiltration stepp needs revision.

Author Response

Dear Reviewer:

According to your comments, a drastic revision of my manuscript has been done. All of the changes have been marked in the revised manuscript in red. The format, spelling, language, information of authors in manuscript have also been revised. The details are as followings.

I marked these changes using “Track Changes” function and colored red.

Reviewer #3:

The authors describe the ability of curcumin to displace alpha-zearalenol from HSA and, thereby, reducing the half-life of alpha-zearalenol and its detrimental effects. The described study is novel and performed well enough. Unfortunately, the manuscript suffers from poor English. As an example, the sentence starting on line 46 "Therefore, we speculated that the interaction between α-ZOL and HSA can be interfered by other substances in vivo circulation, which could reduce the binding constant of α-ZOL and HSA." This sentence should be present tense, not "interfered by" but affected by, and not the binding constant is reduced but the amount of bound α-ZOL. There are a number of more locations that need revision.

Thank you for your suggestions. Apart from the following comments, we have checked and corrected the grammars in our revised manuscript.

  • L69: Should this not be "number of binding sites n" instead of "binding site n"?

Thank you for your comments. The “number of binding sites n” has been revised as “binding site n” in Line 88.

  • L73ff: the abbreviations in the column headers need to be defined. There must be a space between L and mol and a lower case s.

Thank you for your advice. The captions of Table 1 and Table 2 have been revised in Line 93-95 and 140-141.

  • L251: Abbreviations "SC-CO2" and "NSAs" used without definition.

Thank you for your review. We have added the full name of "SC-CO2" and "NSAs" in Line 285-286.

Line 285-286: supercritical carbon dioxide (SC-CO2) technology and nanoporous starch aerogels (NSAs)

  • L272: Sources of the chemicals need to be provided.

Thanks for the suggests, we have added the information of the chemicals in Line 309-314.

Line 309-314: α-Zearalenol (α-ZOL, purity≥98%, Pribolab, Singapore); Curcumin (CUR, purity ≥98%, Victory Biological, China); Human Serum Albumin (Sigma, America); Tris (Bio Basic, America); Warfarin (purity≥97%, Adamas-beta, China) and Ibuprofen (purity≥97%, J&K Scientific, China) are obtained. The water used in this study was all ultrapure water, and all the reagents are analytical grade.

  • L315: Details about the HPLC method to determine free α-ZOL are missing. Description of the ultrafiltration step needs revision.

Thank you for your suggestions. We have made some changes on the ultrafiltration step in Section 5.4 in Line 370-379, and also more details about the HPLC methods were added in Line 387-393.